# Methyl Jasmonate Activates the *2C Methyl-D-erithrytol 2,4-cyclodiphosphate Synthase* Gene and Stimulates Tanshinone Accumulation in *Salvia miltiorrhiza* Solid Callus Cultures

**DOI:** 10.3390/molecules27061772

**Published:** 2022-03-08

**Authors:** Piotr Szymczyk, Grażyna Szymańska, Łukasz Kuźma, Agnieszka Jeleń, Ewa Balcerczak

**Affiliations:** 1Department of Biology and Pharmaceutical Botany, Medical University of Łódź, Muszyńskiego 1, 90-151 Łódź, Poland; lukasz.kuzma@umed.lodz.pl; 2Department of Pharmaceutical Biotechnology, Medical University of Łódź, Muszyńskiego 1, 90-151 Łódź, Poland; grazyna.szymanska@umed.lodz.pl; 3Department of Pharmaceutical Biochemistry and Molecular Diagnostics, Medical University of Łódź, Muszyńskiego 1, 90-151 Łódź, Poland; agnieszka.jelen@umed.lodz.pl (A.J.); ewa.balcerczak@umed.lodz.pl (E.B.)

**Keywords:** callus culture, MEP pathway, promoter, *cis*-active element, tanshinone, methyl jasmonate

## Abstract

The present study characterizes the 5′ regulatory region of the *SmMEC* gene. The isolated fragment is 1559 bp long and consists of a promoter, 5′UTR and 31 nucleotide 5′ fragments of the CDS region. In silico bioinformatic analysis found that the promoter region contains repetitions of many potential *cis*-active elements. *Cis*-active elements associated with the response to methyl jasmonate (MeJa) were identified in the *SmMEC* gene promoter. Co-expression studies combined with earlier transcriptomic research suggest the significant role of MeJa in *SmMEC* gene regulation. These findings were in line with the results of the RT-PCR test showing *SmMEC* gene expression induction after 72 h of MeJa treatment. Biphasic total tanshinone accumulation was observed following treatment of *S. miltiorrhiza* solid callus cultures with 50–500 μM methyl jasmonate, with peaks observed after 10–20 and 50–60 days. An early peak of total tanshinone concentration (0.08%) occurred after 20 days of 100 μM MeJa induction, and a second, much lower one, was observed after 50 days of 50 μM MeJa stimulation (0.04%). The dominant tanshinones were cryptotanshinone (CT) and dihydrotanshinone (DHT). To better understand the inducing effect of MeJa treatment on tanshinone biosynthesis, a search was performed for methyl jasmonate-responsive *cis*-active motifs in the available sequences of gene proximal promoters associated with terpenoid precursor biosynthesis. The results indicate that MeJa has the potential to induce a significant proportion of the presented genes, which is in line with available transcriptomic and RT-PCR data.

## 1. Introduction

In bacteria and plants, two compounds, isopentenyl pyrophosphate (IPP) and dimethylallyl pyrophosphate (DMAPP), produced by the cytosolic mevalonic acid (MVA) and plastidial (MEP) pathways, are used for the biosynthesis of about 40,000 isoprenoids [1]. Their products are used as fragrances, as well as anticancer (Taxol), antimalarial (artemisinin), antithrombotic and antimigraine (ginkgolides) and adaptogenic (ginsenosides) substances [1]. For example, the diterpene-derived tanshinones produced by *Salvia miltiorrhiza* Bunge, also known as red sage or Danshen/Tanshen, have been used in traditional Chinese medicine to treat coronary heart disease. Recently, they have also been used to treat neuropathic pain, alcoholism, hepatic injury, hyperlipidemia, Parkinson’s and Alzheimer’s disease [2,3,4,5]. The annual production of *S. miltiorrhiza* biomass in China alone exceeds 20,000 tons; however, as the approximate yield per hectare is 4.5–6.0 tons, a significant area of arable land is needed to meet this demand [6]. Unfortunately, such land is becoming hard to find due to increasing steppe area, growing soil pollution and water deficit [6].

The enzyme 2C methyl-D-erithrytol 2,4-cyclodiphosphate synthase (MEC) (EC:4.6.1.12) catalyses the fifth step in the seven-step plastidial MEP pathway (7). This is the first and critical cyclization reaction in the MEP pathway converting 4-(cytidine 5′ diphospho)-2C-methyl-D-erithrytol 2-phosphate into 2C-methyl-D-erithrytol 2,4-cyclodiphosphate (MECPD) [7,8]. Although both the MVA and MEP pathways produce IPP and DMAPP, the MVA route provides substrates known to support the biomass growth of *S. miltiorrhiza* hairy roots; the MEP pathway is more responsible for secondary metabolite production, such as biosynthesis of tanshinones in *S. miltiorrhiza* hairy roots [9,10]. Plant hydroxymethylglutaryl-CoA reductase (HMGR) is recognized as the most important enzyme controlling the rate-limiting step in the MVA pathway [11]. The HMGR is precisely regulated in plants at the level of transcription, post-transcription, translation and post-translation [12,13]. Similar rate-limiting function, i.e., the highest metabolite flux control coefficient in the MEP pathway indicates 1-deoxy-D-xylulose-5-phosphate synthase (DXS) [14]. Therefore, the activity of DXS is precisely regulated at several post-translational levels [12,15]. The significance of plastidial MEC enzyme is mediated predominantly by the product of its activity, MECPD, considered to be a retrograde signalling molecule affecting nuclear gene expression [16]. Such hypothesis was verified by studies on mutants of the MEP pathway gene *HDS*, also known as *ceh1* (*constitutively expressing hydroperoxide lyase 1*); these influence the expression of the enzyme HDS (1-hydroxy-2-methyl-2(E)butenyl4-diphosphate synthase) which converts MECPD into hydroxymethylbutenyl diphosphate [16,17,18]. The mutants demonstrated higher MECPD concentrations, elevated salicylic acid (SA) level, greater resistance to infection by biotropic pathogens, and increased expression of a stress-inducible nuclear hydroperoxide lyase gene encoding a plastid-localized protein [16,17,18]. Metabolic engineering approaches in *S. miltiorrhiza* hairy root cultures suggest that the enzyme geranylgeranyldiphosphate synthase (GGPPS) being active at later stages of tanshinone biosynthesis could more strongly induce the tanshinone accumulation rate than HMGR or DXS [19]. The geranylgeranyldiphosphate (GGPP) produced by GGPPS is then used as a substrate by copalyl diphosphate synthase 1 (CPS1) and kaurene synthase-like 1 (KSL1) to the biosynthesis of miltiradiene, representing the complete but biologically inactive carbon structure of tanshinones [20]. Further oxidative modification of miltiradiene skeleton introduced by numerous P450 cytochromes produces biologically active tanshinone molecules [20].

The cDNA of genes coding for *MEC* have been found and characterized in several plants; these fragments usually indicate extensive homology to other plant genes and their expression is positively regulated by light [21,22,23,24]. Information concerning the sequences of *cis*-active regulatory elements in promoter fragments of MEP pathway genes is rather rare; indeed, so far only two 5′ regulatory regions of *Ginkgo biloba isopentenyl diphosphate* synthase (*IDS*) genes- *GbIDS1* and *GbIDS2* have been cloned and characterized [25]. In the case of *Populus trichocarpa*, an initial in silico analysis was performed for all seven MEP pathway genes. Most of the analysed MEP route genes in *P. trichocarpa* have the circadian regulatory motifs CAA(N)_4_ATC and TATTCT; in addition, repeated GATA boxes have been observed, indicating a strong dependence on light induction [21,26].

The plant phytohormone MeJa also participates in the regulation of *MEC* gene expression [27,28]. MeJa facilitates signal transduction in several stages. Firstly, MeJa forms bioactive compounds by complexing with isoleucine [29]. Following detection by CORONATINE INSENSITIVE1 (COI1), COI1 forms complexes with Skp1/Cullin1/F-box proteins (SCF^COI1^) [30]. The Jasmonate protein containing the ZIM domain (JAZ), repressing transcription factor activity, is then ubiquitinated by SCF^COI1^-type E3 ubiquitin ligase and degraded in the 26S proteasome [30]. Removing the JAZ repressor restores the activity of *trans*-factors [31].

A number of transcription factors repressed by JAZ proteins participate in the MeJa-dependent signal transduction route [32]. Some of these are found in the Apetala2/Ethylene-Response Factors, basic Helix-Loop-Helix, WRKY and MYB *trans*-factor families [32].

Most studies related to elicitation of *S. miltiorrhiza* by MeJa have been performed on hairy root cultures; these typically lasted up to 18 days and used MeJa concentrations up to 200 μM [33,34,35,36]. Hou et al. (2021) report that 18-day elicitation with 200 μM MeJa increases the tanshinone concentration in hairy roots of *S. miltiorrhiza* Bunge and *Salvia castanea* f. *tomentosa* Stib [33]. Zhao et al. (2010) also found seven-day MeJa elicitation (10 μM, 50 μM and 100 μM) to increase total tanshinone concentration [34]. In this case, the100 μM MeJa boosted the concentration of CT by 5.0-fold, tanshinone I (TI) by 1.3-fold and tanshinone IIA (TIIA) by 1.4-fold [34]. Hao et al. (2015) report that 100 μM MeJa resulted in a fact induction of total tanshinone concentration in hairy root cultures, with the concentrations reaching a maximum 36 h after treatment and then decreasing stepwise over 144 hrs [35]. Ten-day application of 100 μM MeJa was found to increase CT, TI and TIIA levels. MeJa treatment was also found to demonstrate coaction with exposure to 40 μW/cm^−2^ UVB for 40 min/day [36]. Methyl jasmonate (10, 50 and 100 μM) was also used to stimulate suspension cell cultures of. *S. miltiorrhiza* for 18 days. However, no studies have examined *Salvia miltiorrhiza* solid callus elicitation for longer time-frames, i.e., up to 60 days, and higher MeJa concentrations, up to 500 μM; such long studies have only been used to examine the influence of cytokinin, salicylic acid and auxin on tanshinone biosynthesis rate [37,38].

The present study describes the cloning of a 1559 bp long *S. miltiorrhiza MEC* (*SmMEC*) gene promoter, 5′UTR and a short 5′ CDS DNA sequence (GenBank KT935425.1). Genome Walking method was applied to clone the *SmMEC* promoter. An in silico analysis was performed of the promoter region, which indicated numerous *cis*-active elements; these were validated by comparison with co-expression studies in *A. thaliana*. In the bioinformatic studies was used PlantPAN3.0 and RegSite Plant database [39,40,41]. Moreover, the Bio-Analytic Resource (BAR) was applied to analyse co-expression data [42].

These findings suggested that the *SmMEC* gene is positively regulated by MeJa, which was then verified by RT-PCR analysis. The functional importance of MeJa in the regulation of *SmMEC*, and possibly other tanshinone pathway genes, was assessed during 60-day cultivation of calluses on solid medium containing 50, 100, 250 or 500 μM of MeJa. A significant increase in total tanshinone as well as CT and DHT content was found in the MeJa treated callus cultures compared to untreated controls. Changes in TI, TIIA, CT, DHT and total tanshinone concentration were assessed by a HPLC method. The transcription factors mediated by MeJa play a significant role in tanshinone biosynthesis, as indicated by the distribution of *cis*-elements recognized by MeJa-dependent *trans*-factors in the MEP, MVA and tanshinone-precursor biosynthesis gene promoters. The following databases were examined to obtain promoter sequences: PlantPAN3.0, Arabidopsis org-TAIR, NCBI (Nucleotide) and Uniprot [39,43,44,45,46].

## 2. Results

### 2.1. Isolation of S. Miltiorrhiza MEC Promoter, 5′UTR and 5′fragment of CDS

The *SmMEC* promoter, 5′UTR and a short 5′ fragment of CDS were isolated as described in Materials and Methods. The localization of the transcription initiation site (TIS) was predicted as described in Materials and Methods. The sequence around the TIS at cytidyl 1467 nucleotide (TTA**C**AA, nt 1464–1469), corresponds to a high-scoring TIS sequence (WnT/aC/tA/cw), where W = A or T, as found in 217 dicot promoters (Figure 1) [40]. The TATA-box, usually localized 25–35 bp in the 5′ direction from the TIS, was not observed in the form of cTATAA/TAT/AA described by Shahmuradov et al. (2003) or TCACTATATATAG [40,41,47]. Studies performed on 12749 *A. thaliana* promoters show that only 29% contain TATA motifs clustered around position −32 in relation to TIS [48]. It is possible that the TATA-less promoters could also outnumber those containing TATA-box elements in *S. miltiorrhiza*. The positions of the transcription and translation initiation sites enabled the 5′UTR to be localized (Figure 1).

### 2.2. In Silico Analysis of SmMEC Promoter

The in silico search revealed a lack of tandem repeats and CpG/CpNpG islands. However, potential *cis*-active motifs recognized by *A. thaliana trans*-factors were found in the promoter; these could be homologous to those in *S. miltiorrhiza*. Our findings indicate that light has a significant influence on *SmMEC* gene expression, suggested by the presence of the following light-regulated *trans*-factors; VOZ2, FAR1, CRF2, PIF3 and 5, bZIP68, GBF3, BDOF3.3, LUX, REVEILLE 2,4,5,6,7 and 8, GT1 and HY5. The protection against UV light may be mediated by MYB4. In addition, ER stress and heat shock may regulate *SmMEC* gene expression by a bZIP28 and HSF3 *trans*-factors. *SmMEC* may also respond to such plant hormones as auxin (ARF8, MYB77, MYB93), abscisic acid (ATHB5, MYC2, MYB33, ABF1 and 4, PIF1), brassinosteroids (BIM 1 and 3, BEE2), gibberellin (MYB33, PIF1), ethylene (ERF 3,4,8 and 11, RAV2), elicitors (WRKY14, 15, 17 and 25) and salicylic acid (WRKY70).

A more complex situation could characterize the response to MeJa, that engages different *trans*-factor families such as APETALA2/Ethylene-Response Factors, MYB, WRKY and basic Helix-Loop-Helix (bHLH). The following members of these *trans*-factor families that are previously characterized to participate in plant secondary metabolism are observed in the presented promoter: bHLH (MYC3,4; bHLH14,17), MYB (MYB51,76), WRKY1, ERF1 and 10 [32].

The results of the in silico analysis suggest that *SmMEC* also participates in the response to abiotic stress factors, such as water deficit, salt stress (ATHB12, HDG11, ATHB6 and 7, MYB73 and 74, MYC2, MYB15), or cold (NAC066). Moreover, plant organogenesis may be regulated by changes in *SmMEC* gene expression mediated by YABBY 1,4, SUF4, SPATULA, HAT5 and TCP3, WOX 11 and 13 *trans*-factors. Sample *cis*-active elements localized within the promoter region are presented in Figure 1.

### 2.3. Microarray Co-Expression Studies

The transcription factors and other proteins co-expressed with *A. thaliana MEC* gene (At1g63970; *AtMEC*) were identified by Expression Angler software (Appendix A). The following *A. thaliana* microarray data-set compendiums were used: AtGenExpress Elicitors, AtGenExpress Abiotic Stress, AtGenExpress Chemical Stress, AtGenExpress *Botrytis cinerea,* AtGenExpress *Erysiphe orontii* [42].

Twenty-four *trans*-factor genes were found to be co-expressed with *AtMEC* within the r range 0.7–1.0; these were identified in AtGenExpress Elicitors and AtGenExpress Abiotic Stress (Appendix A). Half of these were found to participate in plant organogenesis and growth regulation (HGD2, bHLH48,60, YABBY1, SPL3,4,5 and 9, SOL1, HBI1, SCAP1, MYB17). These included *trans*-factors engaged in the regulation of circadian rhythm (CP21), response to far-red light (PIF8), cold-stress (ICE1,2) and heat-stress (ERF72). Some co-expressed *trans*-factors could regulate the biosynthesis of flavonol (MYB111), glucosinolate (HB34) and gibberellin (HB25) (File S2), and others of that of phytohormones such as brassinosteroids (BZR2), cytokinins (ARR14), ethylene (ERF72) and abscisic acid (ICE1,HB5,33) (Appendix A).

As it was due to relatively simple to verify their biological effects, particular emphasis was put on plant phytohormones that could regulate *AtMEC* gene activity. However, the analysis of the literature sources suggests that brassinosteroids play a greater role in innate plant immunity and growth process regulation than in the control of secondary metabolism [49]. Moreover, in *S. miltiorrhiza*, ethylene treatment increases tanshinone concentration; however, this level is maintained by members of B-3 subfamily of ERF *trans*-factors, while the co-expressed AtERF72 belongs to the B-2 subfamily [50,51].

Abscisic acid appears to have a particularly interesting influence on the *AtMEC* gene, influencing the activities of three of the 24 identified *trans*-factors in *AtMEC* (Appendix A). A closer review of the literature related to *S. miltiorrhiza* suggests that abscisic acid, similar to polyethylene glycol (PEG), increases the concentration of MeJa in Danshen tissues, with a positive influence on tanshinone biosynthesis rate [52]. These results have driven research towards studying the response of *SmMEC* to MeJa. Initial studies have examined the *r* co-expression value for *trans*-factors MYC4, WRKY1, MYB76, bHLH14 and bHLH17; these are known to participate in the response of *AtMEC* to MeJa [32]. The r-values for these *trans*-factors were lower than 0.7 and varied among used microarray compendiums; the highest values were observed for MYC4 (0.512), WRKY1 (0.568), MYB76 (0.550), bHLH14 (0.378) and bHLH17 (0.209). These figures suggest that the observed co-expressions have relatively low biological relevance for MeJa-dependent *trans*-factors in *A. thaliana*.

These bioinformatic attempts to the role of MeJa in *SmMEC* gene regulation were compared with more significant, transcriptomic studies in *S. miltiorrhiza*. A BLASTX search of results obtained from the Arabidopsis gene regulatory information server (AGRIS) revealed 1377 unique sequences encoding 767 homologous *A. thaliana trans*-factor candidates; of these, 105 were up-regulated and 187 down-regulated [53]. Our in silico test found that four of the ten top up-regulated *trans*-factor genes identified by Luo et al. (2014), including two bHLH (AT1G51070-AtbHLH115, AT4G14410-bHLH104) and two WRKY (AT5G13080-AtWRKY75, AT3G58710-AtWRKY69) genes, demonstrated binding sites within the tested *S. miltiorrhiza MEC* promoter [53]. These results highlight the important role played by MeJa for regulating *SmMEC* activity.

### 2.4. RT-PCR Analysis of SmMDS Promoter Activity after MeJa Administration

RT-PCR analysis revealed decreased *SmMEC* promoter activity after 24 and 48 h of 50, 100 and 250 μM MeJa treatment. A moderate increase in *SmMEC* gene expression was observed only after 72-h MeJa application (Figure 2).

### 2.5. Treatment of S. miltiorrhiza Callus by MeJa

Total tanshinone concentration exhibited generally a biphasic response to elicitation by MeJa. An early peak of total tanshinone concentration occurred after 10–20 days of MeJa induction, and a second, much lower one, was observed after 50–60 days. The initial stimulatory effect was visible as early as after a 10-day treatment with 50 and 100 μM MeJa. Lower concentrations were observed after treatment by 250 and 500 μM MeJa. Total tanshinone concentration increased strongly after 10 days of 50 μM MeJa induction. A higher value was noted for 100 μM MeJa after 20 days of elicitation. The smallest values were noted for 250 and 500 μM MeJa, which also demonstrated lower total tanshinone biosynthesis and accumulation (Figure 3).

Compared to the 20-day value, the total tanshinone content steadily decreased after 30, 40, 50 and 60 days of MeJa stimulation. This trend was observed for all values apart from 50 μM, which showed an increase in total tanshinone content on day 50 as compared to days 20 and 40. After elicitation by 100 and 250 μM MeJa, an increase was noted at day 60 compared to day 50. For 100 and 250 μM, both secondary peaks observed at day 50 and 60 were lower than those after 20 days (Figure 3). It is possible that the secondary increase in total tanshinone observed after 50 and 60 days reflects the dynamic interaction between biosynthesis and degradation of these components in plant tissue and suggests that plants may also increase tanshinone concentration after longer MeJa elicitation periods.

DHT values were found to increase as early as 10 days after MeJa elicitation; the levels peaked after 20 days, with higher values observed for 100 μM than for 50, 250 or 500 μM. A secondary, lower peak of DHT was observed after 50 days of 50 μM MeJa stimulation. No such secondary peaks were observed for other MeJa concentrations (Figure 4).

CT growth generally demonstrated similar kinetics to DHT. Following simulation of 50 μM MeJa, a peak value noted after 10 days. The highest peak value is observed for 100 μM MeJa after 20 days of elicitation, followed by 250 and 500 μM MeJa. Similar to DHT, a secondary, lower peak of CT is observed after 50 days of 50 μM MeJa stimulation (Figure 4).

Both TI and TIIA require much more time as compared to DHT and CT to achieve a peak value. Such peaks were noted after 40–60 days of MeJa elicitation and appear highest when the lower 50 μM MeJa stimulation was applied (Figure 4).

### 2.6. Growth Index Rates of Callus Treated by MeJa

Analysis of GI_F_ rates indicated that both of the highest all MeJa concentrations within the range 50–500 μM inhibited the growth of *S. miltiorrhiza* callus (Figure 5). The strongest inhibition was observed for highest MeJa concentrations. The GI_F_ of 250 μM MeJa was within the range of 0.05–0.14 for the 20- to 60-day period. For 500 μM MeJa treatment, even lower GI_F_ values (<0.02) were observed after 20–40 days, with these values falling to slightly negative values −0.08 over longer periods, suggesting that cell divisions are outweighed by cell death processes. However, our interpretation is not supported by flow cytometry or apoptosis process analysis. Similar results were observed for GI_D_. The higher MeJa concentrations, i.e., 250 and 500 μM, appear to be too high to induce efficient tanshinone biosynthesis by *S. miltiorrhiza* solid callus culture, as demonstrated by the combination of callus growth inhibition with the relatively low total tanshinone and DHT or CT concentration (Figure 5).

Regarding the lower concentrations, the GI_F_ of 100 μM MeJa was about half that noted for 50 μM. Even the lowest MeJa concentration (50 μM) significantly decreased the GI_F_ of *S. miltiorrhiza* callus as compared to the control. The GI_F_ for 50 μM MeJa treated callus ranged from 0.38–1.10 for 20–60 days; in comparison, the controls demonstrated a GI_F_ between 0.84–4.94 while the 100 μM MeJa showed GI_F_ within the range 0.17–0.48 (Figure 5). In addition, 100 μM MeJa treatment yielded approximately twofold (2.39) higher tanshinone concentrations as compared to 50 μM MeJa after 20 days treatment. Although the GI_F_ (0.17) noted that after 20 days for 100 μM MeJa was about a half (0.45) of that observed for 50 μM MeJa (0.38), this is compensated by the 2.39-fold higher total tanshinone concentration. Therefore, the total tanshinone productivity offered by 50 and 100 μM MeJa treated *S. miltiorrhiza* solid callus after 20 days are related.

### 2.7. Distribution of MeJa Responsive Cis-Active Elements within Proximal Promoters of A. thaliana and S. miltiorrhiza

Potential MeJa-responsive *cis*-elements were found in all MEP pathway gene proximal promoters except one: 1-deoxy-D-xylulose-5-phosphate synthase (DXS), 2-C-methyl-D-erythritol 4-phosphate cytidylyltransferase (CMS), 4-hydroxy-3-methylbut-2-enyl diphosphate reductase (HDR), 4-(cytidine 5′-diphospho)-2-C-methyl-D-erythritol kinase (CMK) and 2-C-methyl-D-erythritol 2,4-cyclodiphosphate synthase (MCS). Moreover, three MVA route gene: acetyl-CoA acetyltransferase (AACT), hydroxymethyl glutaryl-CoA synthase (HMGS) and diphosphomevalonate decarboxylase 1 (PMD1) contain such elements (Appendix A).

Furthermore, 15 of the 18 (83.33%) tested *A. thaliana* genes encoding enzymes participating in GGPP biosynthesis indicate an MeJa-responsive *cis*-active motif (Appendix A). In total, among 35 tested *A. thaliana* genes, 25 (71.43%) indicated MeJa-responsive elements (Appendix A). In addition, six of the eight (75.00%) *S. miltiorrhiza* genes were found to contain these *cis*-active motifs within proximal promoters, which may explain the increased tanshinone concentration in response to MeJa treatment (Appendix A).

In addition, closely-spaced MeJa-responsive elements were observed within the *AtIDI* gene (Appendix A). 

The entire pathway of tanshinone biosynthesis is presented on Figure 6 [54,55].

## 3. Discussion

The present study analyses the sequence and biological activity of the *SmMEC* promoter to characterize its potential role in the regulation of gene expression and possible influence on tanshinone biosynthesis rate. The 1559 bp long gene regulatory fragment contains a promoter, 5′UTR and a short 5′CDS segment.

No polypyrimidine tracts were observed in the *SmMEC* 5′UTR segment; however, they were observed in the 5′UTR of the nuclear-encoded spinach chloroplast genes *PsaF* and *PetH* [56]. They are thought to influence the organization of a spliceosomal complex [57,58].

No tandem repeats were observed within the *SmMEC* promoter. Assuming that tandemly repeated DNA sequences indicate a greater propensity to mutate, genes containing tandem repeats in promoters are characterized by higher rates of transcriptional divergence, and this is unlikely to occur in the *SmMEC* promoter [59].

The absence of CpG/CpNpG islands generally suggests a lack of epigenetic, methylation-dependent regulation [60]. As TATA-box regulatory element was observed only in around 29% of *Arabidopsis* promoters, the *SmMEC* promoter represents the majority of TATA-less promoters [48].

Numerous potential *cis*-active motifs recognized by transcription factors were present. Our findings confirm that the *SmMEC* gene promoter participates in plant organogenesis and growth regulation (HGD2, bHLH48,60, YABBY1, SPL3,4,5 and 9, SOL1, HBI1, SCAP1, MYB17). In addition, *trans*-factors engaged in the regulation of circadian rhythm (CP21), response to far-red light (PIF8), cold-stress (ICE1,2) and heat-stress (ERF72) were found. Previous studies suggested that *MEC* gene promoters are induced by light and participate in circadian rhythm regulation [21,22,23,24].

Particular attention was paid to the phytohormone-dependent *trans*-factors observed in *SmMEC*. Although initial bioinformatic tests suggest that *SmMEC* may be regulated by brassinosteroids, ethylene, cytokinins and abscisic acid, more careful literature analysis found MeJa to demonstrate particularly high regulatory potential [32,49,50,51,52].

Although the presented analysis indicates relatively weak co-expression between *SmMEC* and MeJa-dependent *trans*-factors, a functional link between the two has been identified by transcriptomic studies in *S. miltiorrhiza* [53]. Four (AtbHLH104, AtbHLH115, AtWRKY69, AtWRKY75) of ten *S. miltiorrhiza trans*-factors most strongly induced by 12-h MeJa treatment were found to include *cis*-active elements within the *SmMEC* promoter. More importantly, two of these *trans*-factors, AtbHLH104 and AtbHLH115, interact with each other, [61]. Such interactions enable *trans*-factors to build larger dimer- or oligomer complexes with other bHLH or other proteins that resemble their functions during transcription regulation in vivo [61,62,63].

In the present study, *SmMEC* gene expression assessed by RT-PCR initially decreases after 24 and 48 h, with transcription increasing after 72 h of MeJa stimulation. These data are similar to those of Luo et al. (2014), indicating an early inhibitory effect of MeJa on *SmMEC* gene expression after 12 h; however, no such effect was observed by Pei et al. (2018) [53,64]. Although the expression of *SmMEC* gene after MeJa treatment was induced after 72 h, the induction of one gene is not enough to positively affect the entire tanshinone biosynthesis pathway composed of tens of enzymes. 

Our in silico studies applied to proximal promoter regions of MEP, MVA and later stages of tanshinone precursor biosynthesis show that MeJa may regulate the majority of pathway genes. Similar studies of Zhang et al. (2015) were applied to entire promoter regions; however, our findings regarding the concentration of the proximal part may add relevance to obtained results [63]. Moreover, our in silico studies are consistent with results of Pei et al. (2018) showing the broad increase in gene expression among MEP, MVA and later stages of tanshinone biosynthesis after 100 µM MeJa treatment (64). Results of other authors suggest that MeJa has more selective action. Hou et al. (2021) report increased expression of *HMGR*, *HMGS*, *MK*, *PMK*, *MDC* and *AACT*, while Luo et al. (2014) show that only selected genes as *DXS*, *HMGR*, *MK*, *PMK*, *IPPI* were stimulated by MeJa [33,53]. It is possible that the observed differences may be due to the fact that Luo et al. (2014) only used 12-h transcriptomic studies, Pei et al. (2018) used six-day studies [53,64].

Genome-wide transcriptomic studies in *S. miltiorrhiza* confirm that MeJa positively regulates the *DXS*, *HDR*, *CMK* and *MCS* genes in the MEP pathway, as well as the *AACT*, *PMD* and *hydroxymethylglutaryl-CoA reductase* (*HMGR*) genes in the MVA pathway [20,53,54,55]. In addition, the later stages of the terpenoid biosynthesis pathway also appear to depend on MeJa regulation, as reflected in the induction of the isopentenyl-diphosphate delta-isomerase (IDI), farnesyl diphosphate synthase (FPPS), geranyl diphosphate synthase (GPPS) and geranylgeranyldiphosphate synthase (GGPPS) genes [20,53,54]. These data suggest that also the genes of *SmKSL1* and *SmCPS1* synthases are stimulated by MeJa [20,53,54,55]. Furthermore, RT-PCR examination showed that *SmDXS*, *SmHDR*, *SmCMK*, *SmIDI*, *SmFPPS*, *SmGGPPS*, *SmCPS* and *SmKSL1* were positively regulated by MeJa [10,20,35,36,54]. Finally, our present RT-PCR findings support the induction of the *S. miltiorrhiza 2-C-methyl-D-erythritol 2,4-cyclodiphosphate* synthase gene (*SmMCS*) by MeJa after 72 h; in contrast, shorter 24- and 48-h exposure to MeJa inhibits *SmMEC* activity. In addition, closely-spaced MeJa-responsive elements were observed within the *AtIDI* gene. Such closely-spaced repetitions of *cis*-active motifs enables the formation of homodimers of *trans*-factors, as observed in in vivo conditions [61,62,63,64,65].

Assuming that MeJa can activate numerous isoprenoid biosynthesis *SmMEC* gene expression after 72-h treatment, this confirms the hypothesis that it is able to increase the biosynthesis rate of tanshinone in *S. miltiorrhiza* solid callus culture. As the enzymes participating in late stages of tanshinone biosynthesis, such as SmCPS1, SmKSL1 and SmCYP76AH1 are expressed in roots of *S. miltiorrhiza*, most research related to *S. miltiorrhiza* MeJa elicitation has been performed on hairy or callus cultures [35,52,65,66]. These studies show that tanshinone accumulation in *S. miltiorrhiza* hairy roots generally occurs within a few days [33,34,35,64]. These studies tested the *Salvia miltiorrhiza* hairy root response to MeJa using relatively low phytohormone concentration of up to 200 µM in the short duration of up to 18 days. Therefore, a gap exists in available research regarding *Salvia miltiorrhiza* solid callus elicitation using longer time-frames up to 60 days and higher MeJa concentrations up to 500 μM.

Although results of other authors were performed on hairy roots and not callus, they are generally in line with our presented findings, showing an accumulation of total tanshinones beginning after 10 days of elicitation and achieving max values after 20 days. The highest values were observed for 100 μM MeJa. However, the 50 μM, 250 and 500 μM MeJa induced lower total tanshinone concentrations, with both highest MeJa concentrations showing strong callus growth inhibition. After 30–60 days, a decrease in total tanshinone concentration was observed. An interesting exemption was 50, 100 and 250 μM of MeJa, which yielded a secondary tanshinone concentration increase after 50 and 60 days. It is possible that the highest 500 μM MeJa concentration direct the metabolic processes too strongly toward secondary routes, resulting in slower primary metabolism and decreased tanshinone accumulation rate. The more balanced metabolism stimulated by 50 and 100 μM MeJa results in a clear secondary tanshinone peak; this may be a result of slow accumulation reflecting the dynamics between the synthesis of tanshinone, their processing into other tanshinones, degradation or diffusion into the culture medium [53,65].

Previous studies indicate low tanshinone concentrations (0.09–0.40 mg/g DW) within *S. miltiorrhiza* callus developed from leaf, stem and petiole [36,67]. Gryszczynska et al. (2015) also report a generally low total tanshinone concentration (0.077%) in *S. miltiorrhiza* callus, which is significantly less than roots regenerated in vitro (0.269–1.137%) or in native plant roots growing in China (0.260–0.388%) [2,68,69]. A significant factor contributing to generally low tanshinone concentration may be the lack of rhizome, being the place of tanshinone accumulation or the cultivar type, as *S. miltiorrhiza* plants in Poland are characterized by a tanshinone concentration of 0.01–0.26% [70].

Our findings indicate that the maximal tanshinone concentrations achieved in *S. miltiorrhiza* solid callus cultures after 20 days of 100 μM MeJa elicitation are 0.08%.

The dominant tanshinones induced by MeJa elicitation are CT and DHT, with their amounts peaking after 20 days for 100 μM MeJa, as also noted by Hao et al. (2013) and Pei et al. (2018) [10,64]. In addition, CT and DHT levels clearly increased from as early as 10 days after; this observation is in line with previous studies performed on *S. miltiorrhiza* hairy roots [34,35,36]. The dominant position of CT and DHT as well as relatively prompt accumulation may result from their localization in the biosynthesis pathway, where CT is further modified to TIIA, and DHT is used to produce TI [20]. Therefore, the TI and TIIA require much more time within 40–60 days to achieve their peak values.

The imbalance in the callus primary and secondary metabolic pathways induced by the 250 and 500 μM MeJa concentrations may be reflected in the fact that the decrease in callus GI_F_ is concentration-dependent [53,64]. Over the 20–60 day period, the GI_F_ value was within the range of 0.05–0.14 for 250 μM MeJa, and negative (−0.08) at the higher MeJa concentration (500 μM). Similar negative values of GI_F_ (−16%) were observed in *Salvia castanea* hairy root cultures growing for seven days on 300 μM MeJa [71]. However, Li et al. (2020) report that even higher MeJa concentrations (400 μM) stimulate the growth of *Salvia przewalskii* Maxim. hairy roots [72].

Our findings indicate that even the lowest MeJa concentration (50 μM) significantly inhibited the GI_F_ by 0.38–1.10 for 20–60 days as compared to controls (GI_F_ 0.84–4.94). Assuming that the GI_F_ for 100 μM MeJa is approximately half that of 50 μM, and the concentration of tanshinones in callus growing on 100 μM MeJa is only 2.39-fold higher than on 50 μM MeJa, the optimal concentration that could be used for *S. miltiorrhiza* solid callus elicitation is 50–100 μM.

The relationship between the transcription rate of MEP pathway enzymes and concentration of corresponding proteins was evaluated in *S. miltiorhiza* hairy root cultures elicited by 1 g L^−1^ yeast extract and 0.41 mM Ag^+^ [73]. Quantitative LC-MS/MS analysis was applied to test precisely the protein concentration in the presented system. Obtained results suggest, that three MEP pathway proteins: 1-deoxy-D-xylulose-5-phosphate synthase (DXS), 2-C-methyl-D-erythritol 4-phosphate cytidylyltransferase (CMS), and 2C methyl-D-erithrytol 2,4-cyclodiphosphate synthase (MEC) were significantly upregulated after two weeks of elicitation. Extending the elicitation time to five weeks decreased or completely ceased the initial upregulation [73]. The overexpression of *MDS* alone in *S. miltiorrhiza* hairy roots increases not only the concentration of the corresponding mRNA assayed by RT-PCR but also induces the total tanshinone concentration evaluated by HPLC [74]. Presented data support the presence of putative relationship between increased *MDS* gene expression and upregulation of MDS enzyme levels. Related proteomic studies were not performed on *S. miltiorrhiza* plant material treated by MeJa and could be addressed in future research.

## 4. Material and Methods

### 4.1. Plant Cultivation

Seeds used for cultivation of *S. miltiorrhiza* Bunge were obtained from Medicinal Garden of the Department of Pharmacognosy at the Faculty of Pharmacy, Medical University of Lodz (Łódź, Poland). Plants were grown on composite soil, in 0.5 L pots (diameter 12 cm), at 26 ± 2 °C under natural light. Plants of eight weeks old were used for experiments.

### 4.2. Isolation of Genomic DNA

The genomic DNA was prepared according to Khan et al. (2007) [75]. Approximately 0.75 g of plant material was used. The genomic DNA concentration and purity was determined based on A_260/280_ and A_260/230_ using a P300 Nanophotometer (Implen, Munich, Germany).

### 4.3. Promoter Isolation and In Silico Characterization

The hypothesized promoter region of *S. miltiorrhiza MEC* gene (*SmMEC*) was isolated using a Genome Walker^TM^ Universal kit (Takara Bio USA, Mountain View, CA, USA). The 5′ terminal fragment of the *SmMEC* cDNA fragment (GenBank JN831097.1) was used to design two specific primers GSP1 5′ATCAAGACCTCCTCAAAGCAACCCACCGT3′ and GSP2 5′TATGGCTGGTTCCCTCTGCTACGCCG’3. The procedure was performed according to the instructions of the Genome Walker^TM^ Universal kit manufacturer. The length of GSP1 and GSP2 was extended up to approximately 26–29 nt, allowing a high salt-adjusted (50 mM NaCl) melting temperature for both primers (T_M_), i.e., within 67–70 °C; this enables better DNA replication specificity during a PCR reaction. The annealing and extension PCR steps were performed at the same temperature (67–72 °C) according to the manufacturer’s instructions. Both gene-specific GSP1 and GSP2, as well as the DNA sequencing primers, were designed with the help of OligoCalc server [76]. The salt adjusted (50 mM NaCl) T_M_ was used to characterize T_M_ of all PCR primers.

A hot-start type Advantage^®^ 2 Polymerase Mix (Takara Bio USA, Mountain View, CA, USA) was used to further increase the specificity of the PCR reaction. The primary and nested PCR reactions were performed in the following conditions. The first PCR reaction was performed thus: initial denaturation (95 °C, 3 min), denaturation (95 °C, 30 s), annealing and extension (72 °C, 3 min) for seven cycles. The next 32 cycles comprised denaturation (95 °C, 30 s), annealing and extension (67 °C, 3 min). Finally, an additional cycle (67 °C, 7 min) was performed. The nested PCR reaction was performed thus: initial denaturation (95 °C, 3 min), denaturation (95 °C, 30 s), annealing and extension (72 °C, 3 min) for five cycles. The next twenty cycles comprised denaturation (95 °C, 30 s), annealing and extension (67 °C, 3 min). Finally, an additional cycle (67 °C, 7 min) was performed.

The products of the PCR reaction were TOPO-TA cloned into pCR^®^-II TOPO^®^ vector (Thermo Fisher Scientific, Waltham, MA, USA) according to the manufacturer’s instructions. The plasmid DNA was isolated by alkaline lysis miniprep and purified by phenol-chloroform extraction [77]. The purity of plasmid DNA was assessed based on A_260/280_ and A_260/230_ using a P300 Nanophotometer (Implen, Munich, Germany). Plasmid DNA was sequenced (CoreLab facility, Medical University of Lodz, Poland). The obtained DNA sequence was deposited at GenBank (KT935425.1) and analysed by a PlantPAN3.0 to find potential *cis*-active sequences, tandem repeats, miRNA binding sites and CpG/CpNpG islands [39]. TATA-box and transcription initiation site (TIS) were identified using TSSP software and RegSite Plant DB (Softberry Inc., Mount Kisco, NY, USA) [39,40,41,78,79].

### 4.4. Microarray Co-Expression Studies

The in silico analyses of the *SmMEC* promoter were validated based on the results of *Arabidopsis thaliana* co-expression studies: the *trans*-factors identified by the in silico searches of the *SmMEC* promoter region were compared with those co-expressed with the *A. thaliana MEC* gene (At1g63970; *AtMEC*). Co-expression analysis was performed as described in earlier research [78]. Only values of r within the range 0.7–1.0 were selected, as proposed by Usadel et al. [80].

Given the relatively large number of individual samples studied in each microarray or RNA sequencing test, even co-expression rates (r) of 0.2 are statistically significant, especially when multiple arrays are used to calculate r values. Formally, the values of statistical significance (P) may be calculated from the r rates by Excel using the following equation:P = TDIST{ABS[r/SQRT({1 − r ∗ r}/{n − 2})],[n − 2],2}. (1)

The number of samples is indicated by n. However, as the obtained co-expressions rates (r) of 0.2 could be biologically irrelevant, a purely statistical approach based on *p*-values may give an incorrect outcome. To ensure that the obtained co-expression results are biologically relevant, only values of r within the range 0.7–1.0 were selected, as proposed by Usadel et al. (2009) [80]. The Bio-Analytic Resource (BAR) developed at University of Toronto (Canada) at was used to analyse co-expression data [42].

### 4.5. Callus Induction and Methyl Jasmonate (MeJa) Treatment

The callus applied in the study was developed according to Wu et al. (2003) with modifications comprising a two-fold increase in leaf explant size and cultivation of callus in a 100 mm diameter glass Petri dishes instead of the original 22 mm × 160 mm glass tubes [37]. Therefore, 10 mm × 10 mm leaf fragments from the *S. miltiorrhiza* plants were used as explants for the induction of callus. The explants were surface disinfected with 70% ethanol for 30 s, followed by treatment with 0.5% sodium hypochlorite for 10 min. The excess sodium hypochlorite was removed by fivefold rinsing with sterile distilled water. Explants were cultured in a 100 mm diameter glass Petri dishes containing 20 mL of Murashige and Skoog’s (MS) basal medium supplemented with 3% sucrose, 1% Difco Bacto agar (Difco Laboratories, Detroit, MI, USA) and 1 mg L^−1^ 2,4-dichlorophenoxyacetic acid (2,4-D) [81]. To ensure that no microbiological contamination was present, the Petri dishes with the explants were sealed with two layers of Parafilm (Pechiney Plastic Packaging, Chicago, IL, USA). Moreover, to reduce the risk of accidental microbiological contamination, all works were performed under laminar hood. The cultures were incubated at 26 ± 2 °C in a darkness for a period of one month. The samples of callus were then subcultured in the same conditions for one month to increase the mass of callus. Callus samples (approximately 1000 mg) were then cultured on Murashige and Skoog (MS) basal medium supplemented with 3% sucrose, 1% Difco Bacto agar (Difco Laboratories, Detroit, MI, USA) and either 50, 100, 250 or 500 μM of MeJa. The callus cultures were maintained for up to 60 days. Every 20 days, fresh solid medium was provided. The control group was the MeJa untreated callus.

A stock solution of MeJa (50 mM) in 70% ethanol was sterilized by a syringe filter (0.4 μm pore size) under a laminar hood before addition to warm (50 °C), freshly-autoclaved MS medium under a laminar hood. The pH of each medium before adding MeJa was adjusted to 5.7 ± 0.1 with 1N NaOH or HCl. Following this, the media were autoclaved for 20 min at 121 °C, 105kPa. The glass and steel forceps used to manipulate with calluses were sterilized at 200 °C for 1 h before entering the laminar hood.

### 4.6. Calculation of Callus Growth Index

Callus samples for growth index calculation were collected after 20, 40 and 60 days. All the experiments were repeated three times, and growth measurements were performed with three replicates per harvesting. Callus untreated with MeJa was used as a control group.

To protect against microbiological contamination of the callus, its mass was determined on an electronic scale placed under a laminar hood. The electronic scale and working surface of the laminar hood were initially sterilized with UV irradiation (30 min) and then with 70% ethanol. The glass material and steel forceps used to manipulate the calluses were sterilized for 30 min at 200 °C before use in the laminar hood. Plastic tips were autoclaved as liquid media.

Growth index (GI) was calculated according to Godoy-Hernández and Vázquez-Flota (2006) [82]. The GI of callus fresh weight was calculated as follows:GI_F_ = (FW_F_ − FW_I_)/FW_I_.(2)
where: GI_F_ = GI of callus fresh weight; FW_F_ = final callus fresh weight FW_I_ = initial callus fresh weight.

The same approach was used to calculate the growth index of dry callus (GI_D_). Callus samples for GI_D_ were freeze-dried as described above.

### 4.7. Extraction of Callus

Callus samples for HPLC experiments were harvested every 10 days for up to 60 days. The samples harvested after 10 days were grown on the same medium they started growth, while those grown longer than 20 days were transferred to new, fresh medium. The harvested callus samples were freeze-dried in an Alpha 1–2 LD lyophilizer (Martin Christ, Osterlode, Germany). The 30 mg dry callus was finely ground with mortar and pestle and extracted with 1.5 mL methanol under 60 min sonication (UM1 disintegrator, Unimal, Olsztyn, Poland) at room temperature. The extraction was performed according to Wan et al. (2009) [83].

Methanol or 80% methanol solution is commonly used to extract tanshinones from *S. miltiorrhiza* plant material [73,84,85]. This approach avoids the need for low pressure evaporation of additional solvents such as chloroform and re-solubilization of tanshinones in methanol before beginning HPLC [86]. This is particularly valuable in experiments based on small amounts of dried plant material which may only measure tens of mg [83]; such samples only produce tiny amounts of extracted tanshinones and these may not be completely solubilized in the added methanol, resulting in relatively high error rates.

The samples were then centrifuged at 12,000× rpm for 10 min at room temperature. Samples were filtered through 0.45 μm Chromafil membrane (Machery-Nagel, Duren, Germany) and used for HPLC analysis. Samples were used instantly for HPLC analysis or stored at −25 °C in dark glass vials to avoid tanshinone decomposition.

### 4.8. HPLC Analysis

All analyses were performed on Agilent 1200 HPLC System (Agilent, Palo Alto, CA, USA) equipped with an auto sampler, quaternary-pump delivery system, on-line degasser, column temperature controller and UV-VIS DAD detector. The system was connected with Agilent ChemStation 2001–2010 software. Chromatographic separation was performed on the Agilent Zorbax Extend C18 reversed phase column (5 μm, 250 mm × 4.6 mm) with an Agilent Zorbax Extend C18 guard column (5 μm, 10 mm × 4.6 mm). The detection was accomplished at 270 nm, the time of analysis was 20 min, the flow rate was 1.2 mL min^−1^ and the column temperature was maintained at 20 °C, the sample volume was 20 μL. The mobile phase consisted of A (water for HPLC) and B (acetonitrile). The following gradient program was applied: initially 45% B at 0 min, linearly increasing to 60% B at 2 min, maintaining 60% B from 2 min to 9 min, linearly augmenting B to 80% at 10 min, linearly expanding B to 82% at 13 min and finally linearly decreasing B to 45% ant 20 min. After each analysis, 45% B was pumped and held for 10 min to re-equilibrate the system for baseline stability. The procedure is a modified method of Liu et al. (2006) [86].

Standard HPLC-grade substances such as TI, TIIA, CT and DHT were provided by Sigma Aldrich Poland (Poznań, Poland). Methanol, acetonitrile and water for HPLC (J.T. Baker HPLC Analyzed) were received from Avantor Performance Materials (Gliwice, Poland).

### 4.9. Preparation of Calibration Standard Solutions

Stock methanolic solutions of 0.1 mg L^−1^ TI, TIIA, CT and DHT were prepared for instant use or were stored in the dark at −25 °C. These initial standard solutions were used to prepare standard curves. Each calibration curve was analysed three times with six different concentrations using the same HPLC conditions as described earlier.

The DHT indicated a retention time of 7.1 min with a calibration curve of y = 2964.761x + 1.1061 and a correlation coefficient of 0.9999. The CT showed a retention time of 10.8 min, with a calibration curve of y = 4291.427x − 0.4089 and a correlation coefficient of 0.9995. The TI indicated a retention time 11.6 min, calibration curve y = 2450.131x − 1.8135 and correlation coefficient 0.9996. The TIIA showed a retention time 13.7 min, standard curve y = 5323.0061x + 0.9614 and a correlation coefficient 0.9998. The linearity range of all calibration curves is 1.25–50 ng.

### 4.10. RNA Isolation and cDNA Synthesis

The RNA was prepared from *S. miltiorrhiza* callus using an Isolate Plant II RNA kit (Bioline, Singapore) according to the manufacturer’s instructions. Briefly, 80–100 mg samples of plant leaves were cut off and frozen straight away in liquid nitrogen. The samples of RNA were digested by RNase-free DNaseI (4 U/sample) to ensure the complete removal of genomic DNA. The expected result of DNaseI digestion was determined by the quantitative, real-time PCR reaction using control samples of RNA, without the standard reverse transcription step. All types of RNA samples were prepared in triplicate. All RNA samples were processed instantly or stored at −80 °C until analysis. The concentration and purity of the prepared RNA was evaluated using a p300 Nanophotometer (Implen, Munich, Germany). The A_260/280_ ratio of isolated RNA was within the range of 1.6–1.8.

The obtained RNA was used as a substrate in the reverse transcription reaction using an Enhanced Avian HS RT-PCR Kit (Sigma-Aldrich, Poznań, Poland). The reaction mixture contained the following components: dNTPs (1 mM final), anchored oligo (dT)23 (3.5 μM final), 2 μL of 10× buffer, RNase inhibitor (20 U), an Enhanced Avian Reverse Transcriptase (RT; 20 U). The quantity of RNA was adjusted to achieve a final RNA concentration of 0.01 μg/μL in a final volume of 20 μL.

### 4.11. Real-Time PCR

Real-time PCR was used to analyse the relative concentrations of *MEC* and *ubiquitin* mRNA in *Salvia miltiorrhiza* callus samples. The Rotor-Gene 6000 (Corbet) and SYBR Green Jump Start Tag ReadyMix™ (Sigma-Aldrich, Poznań, Poland) were used in the RT-PCR tests. Experiments were performed in duplicate to ensure the reproducibility of the method.

*Ubiquitin* was chosen as a reference gene due to its very stable expression in *S. miltiorrhiza* [87]. This was confirmed by analysis with bestKeeper software, which indicated low SD (SD < 1) and relatively high r values [88].

Ethidium-bromide agarose gel electrophoresis and sequence alignment confirmed the expected size of the *ubiquitin* gene fragment (192 bp) based on *Populus trichocarpa* cDNA (GenBank FJ438462.1) using primers 5′GTTGATTTTTGCTGGGAAGC3′ (forward) and 5′GATCTTGGCCTTCACGTTGT3′ (reverse) [78,79]. Similarly, the *S. miltiorrhiza MEC* gene fragment (GenBank JN831097.1) was found to be 117 bp in length based on the *SmMEC* gene primers 5′GGCTGGTTCCCTCTGCTA3′ (forward) and 5′ACGAGGGAAGCTGCAAGTTT3′ (reverse).

The RT-PCR reactions were performed in separate tubes. Samples and negative controls were created in triplicate. The following qPCR reaction parameters were used: initial denaturation (95 °C, 10 min), denaturation (95 °C, 20 s), primer annealing (60 °C, 30 s), extension (72 °C, 20 s). In total, 40 PCR cycles were performed. The following components were added to the reaction mixture: 7.5 μL SYBR-Green ReadyMix, 0.7 μL of each primer, 1 μL of cDNA and distilled water to a final volume of 16 μL. The equation of the standard curve was y = −3.29 − 7.2598, R^2^ = 1. The relative changes in gene expression were calculated according to the 2−ΔΔ*C*T method developed by Livak and Schmittgen (2001) [89,90]. The qPCR results were analysed by Rotor-Gene 6000 Series Software 1.7 (Qiagen, Hilden, Germany). 

### 4.12. Statistical Analysis

The results of the HPLC. GI and RT-PCR analysis were evaluated by the Kruskal–Wallis test. Statistical analysis was performed using STATISTICA (StatSoft Inc. 2013, version 13.1). The Wilcoxon signed-rank test was used to test samples before and after MeJa treatment with the aim of calculating p values. The Wilcoxon signed-rank test does not assume that the differences between paired samples are normally distributed. Therefore, the Wilcoxon signed-rank test has greater statistical power than Student’s *t*-test and is more likely to produce a statistically significant result. Values of *p* < 0.05 were considered statistically significant.

### 4.13. Promoter Analysis of MEP, MVA and Tanshinone Precursor Biosynthesis Genes

To give a more accurate picture of the MeJa induction of tanshinone biosynthesis in *S. miltiorrhiza* solid callus cultures, the distribution of the *cis*-active elements was analysed within the proximal promoters of MEP and MVA and the later stages of tanshinone precursor biosynthesis genes in *A. thaliana* and *S. miltiorrhiza* [38]. The obtained data were compared with available transcriptomic studies of *S. miltiorrhiza* leaf genes induced by 12-h MeJa treatment [53].

The promoter regions of *A. thaliana* can be up to 1 kb in length [91]. However, the present searches were concentrated on proximal promoters within 300 bp of the transcription start site. Such a high concentration on proximal promoters ensures that the *cis*-active motifs found within these regions have higher biological relevance [92,93,94].

The following databases were examined to obtain promoter sequences: PlantPAN3.0, Arabidopsis org-TAIR, NCBI (Nucleotide) and Uniprot [39,43,44,45,46]. The transcription start sites in *S. miltiorrhiza* promoters were characterized using TSSP software [41].

MeJa is known to activate the following *trans*-factor families: Apetala2/Ethylene-Response Factors, basic Helix-Loop-Helix, WRKY and MYB (32). Therefore, the selected gene proximal promoters were searched for the following *cis*-active elements: GCC-box (AGCCGCC) bound by Ap2/ERF *trans*-factors, W-boxes TTGAC(C/T) recognized by WRKYs, the R2R3-MYBs protein DNA binding sequence AACNGC, E-box (CANNTG) and its variant G-box (CACGTG) that associate with bHLHs [32,95,96,97,98,99,100,101].

## 5. Conclusions

The present paper characterises the 5′ regulatory region of the *SmMEC* gene. The *SmMEC* promoter region contains repetitions of many potential *cis*-active elements serving as the recognition sites for transcription factors. These observations are verified by co-expression studies based on *A. thaliana* microarray data and available references.

Our findings confirmed the presence of *cis*-active elements associated with response to methyl jasmonate in the *SmMEC* gene promoter. The response to methyl jasmonate was also confirmed by RT-PCR tests.

Treatment of *S. miltiorrhiza* solid callus cultures by 50–500 μM MeJa indicated a biphasic total tanshinone accumulation kinetics, with peaks observed after 10–20 and 50–60 days for the 50, 100 and 250 μM MeJa concentrations. The dominant tanshinones induced by MeJa are CT and DHT.

To better characterize the effect of MeJa treatment on tanshinone biosynthesis, the sequences of the gene proximal promoters associated with terpenoid precursor biosynthesis (MEP, MVA, GGPP) were searched to find methyl jasmonate-responsive *cis*-active motifs. The same test applied to available promoter sequences of *S. miltiorrhiza* genes indicated that MeJa was able to induce a significant part of the tested genes.

MeJa stimulation inhibits *S. miltiorrhiza* solid callus growth in a concentration-dependent manner.

## Figures and Tables

**Figure 1 molecules-27-01772-f001:**
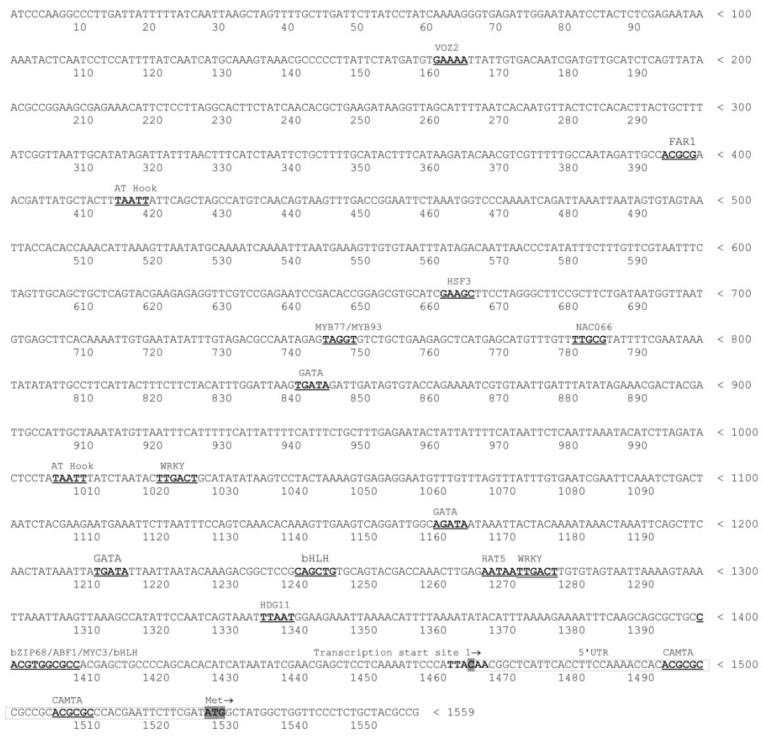
Sequence of *S. miltiorrhiza MEC* promoter region (nt 1–1559), 5′UTR (nt 1467–1528), and 5′ fragment of CPS cDNA (nt 1529–1559). Only strand + is provided. Positions of *cis*-active elements were underlined and bolded. *MEC* gene 5′UTR is underlined. Transcription start site at C in the position 1467 and initial Met ATG codon are shaded.

**Figure 2 molecules-27-01772-f002:**
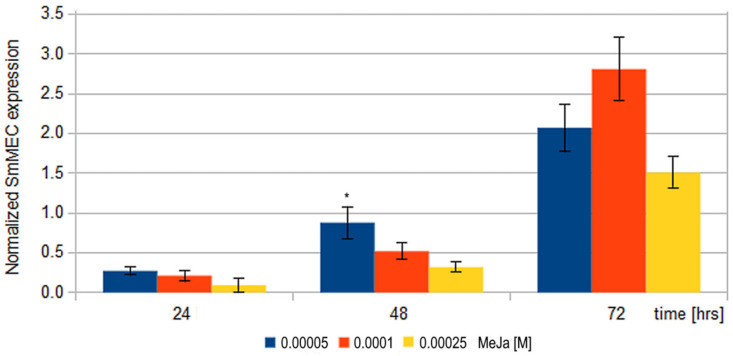
Temporal changes of *SmMEC* gene expression evaluated at 24, 48, and 72 h after treatment by 50, 100 and 250 μM MeJa. Results presented as normalized *SmMEC* expression. Results not statistically significant (*p* > 0.05) were marked by asterisk (*), all other results were statistically significant *p* < 0.05.

**Figure 3 molecules-27-01772-f003:**
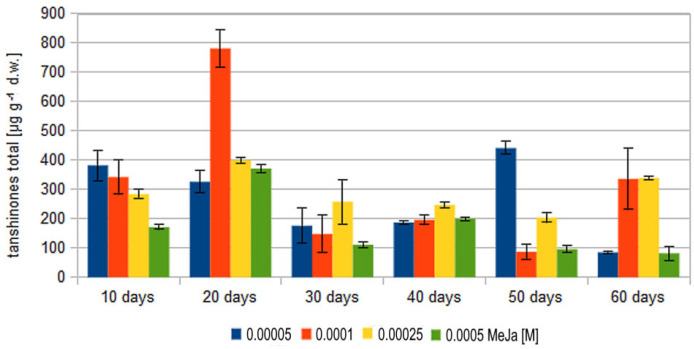
Concentration of total tanshinone as a function of methyl jasmonate (MeJa) concentration (50, 100, 250 and 500 µM) and elicitation time 10–60 days. Control values were not presented as tanshinone concentration was not detectable. All results were statistically significant *p* < 0.05.

**Figure 4 molecules-27-01772-f004:**
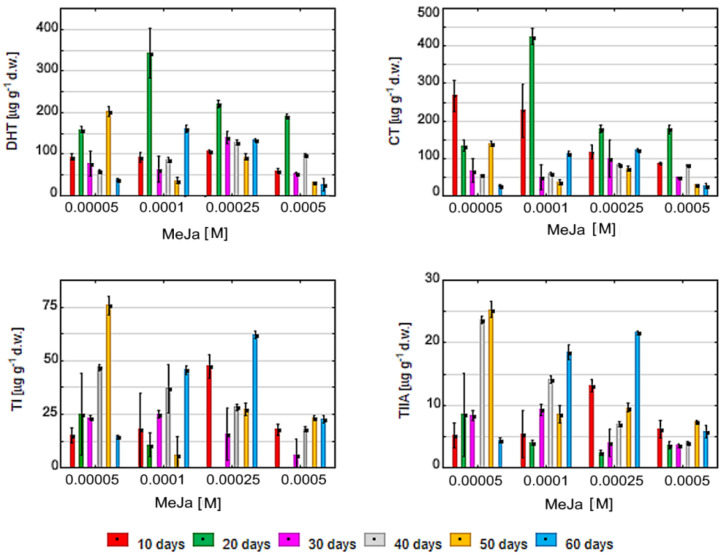
Concentration of four tanshinones CT, DHT, TI and TIIA in *S. miltiorrhiza* callus presented as a function of methyl jasmonate (MeJa) concentration (50, 100, 250 and 500 µM) and elicitation time 10–60 days. Control values were not presented as tanshinone concentration was not detectable. All results were statistically significant *p* < 0.05.

**Figure 5 molecules-27-01772-f005:**
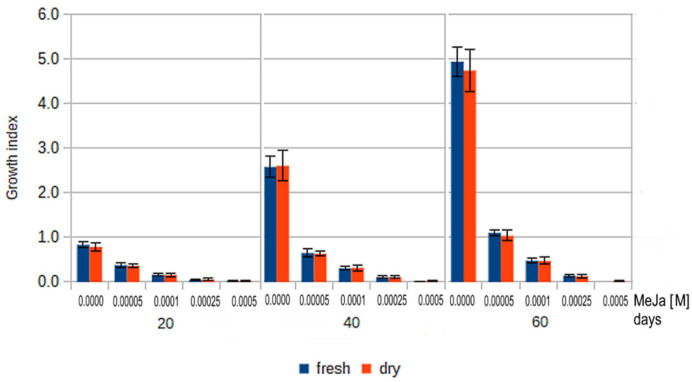
Growth index for fresh (blue) and dry (orange) *S. miltiorrhiza* callus presented for control and four MeJa concentrations 50, 100, 250 and 500 µM. All results were statistically significant *p* < 0.01.

**Figure 6 molecules-27-01772-f006:**
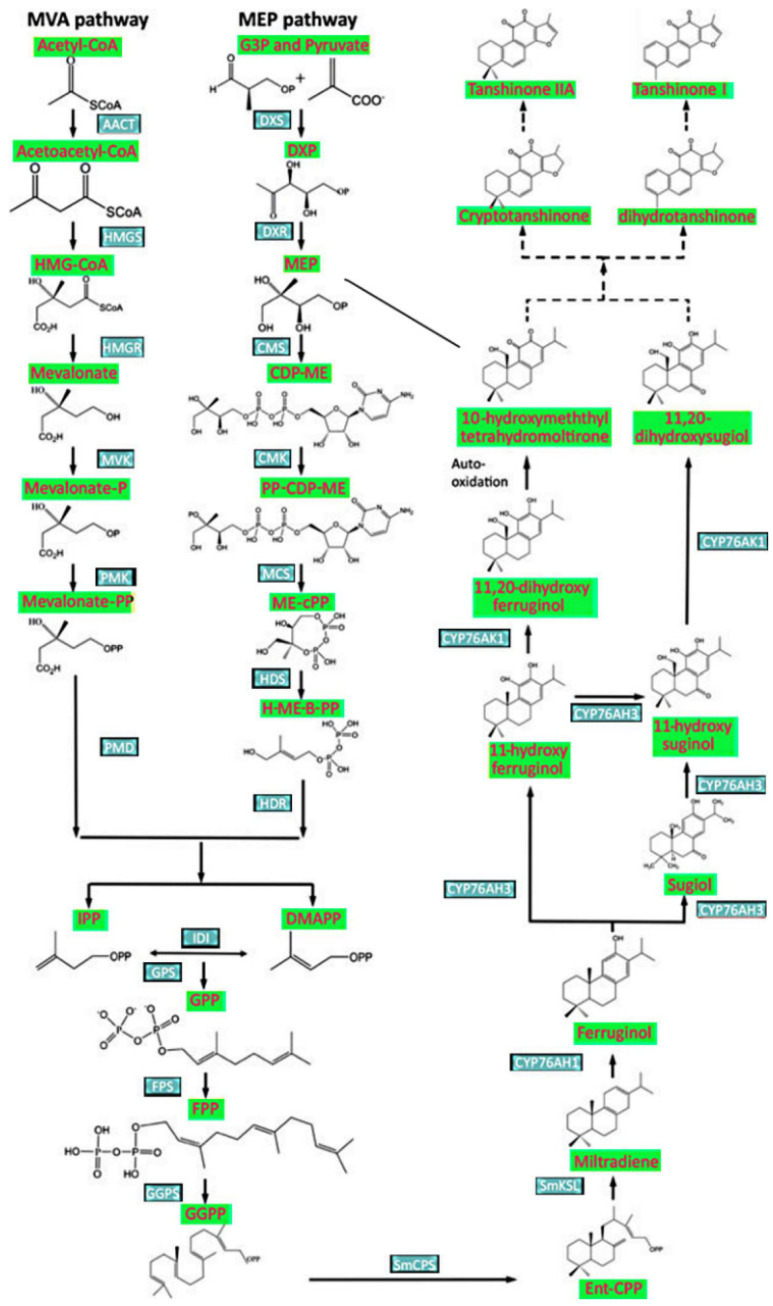
Enzymatic reactions participating in tanshinone biosynthesis [54,55].

## Data Availability

Not applicable.

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
