# Peer review of "Methyl Jasmonate Activates the 2C Methyl-D-erithrytol 2,4-cyclodiphosphate Synthase Gene and Stimulates Tanshinone Accumulation in Salvia miltiorrhiza Solid Callus Cultures"

_molecules, 2022, doi:10.3390/molecules27061772_

Round 1
Reviewer 1 Report
The present paper characterizes the 5’ regulatory region of the SmMEC gene. The content sound good and here are my recommendations:
The Abstract is too wordy and needs to be reframed by adding some value eg % of the callus. Also in the introduction, the clear methodology used to achieve the result is missing. In the Introduction section, Many sentences are without reference kindly add references and correct this. Present clearly the objective of the work. Results are well presented however, figures 2 -5 should be improved by adding the pvalue to the different groups check some packages in R for the analysis.
The second point to address is to add the gene pathway for the main gene to the result section. Describe very well the different possible mutations through a clear figure. The discussion needs to be reframed as well to meet to journal requirements. In the materials and methods, the statical analysis is very weak and authors must improve it. The English language needs to be edited as well across the document.
Author Response
Answers to Reviewer comments
Reviewer 1
We thank the Reviewer 1 for comments and the opportunity to improve the manuscript.
We carefully analyzed Reviewer 1 comments and provide following answers.
- The Abstract is too wordy and needs to be reframed by adding some value eg % of the callus.
The Abstract has been shortened- sentences to be removed are marked in red. Now the Abstract has less than 200 words. A sentence was added (marked in green) describing the concentration (%) of tanshinones in callus culture. Sentences to be removed are marked in red.
- In the introduction, the clear methodology used to achieve the result is missing.
In the introduction section, several sentences were included to better describe the used methodology. These sentences are marked in green. Sentences to be removed are marked in red.
- In the Introduction section, many sentences are without reference.
References were added to the sentences in the Introduction section. Several novel references were added.
- Figures 2 -5 should be improved by adding the p value
Values of p were calculated using Wilcoxon signed-rank test. We added the calculated p value to the description of Figures 2-5 to make them more clear. We added short information of this test to the Material and method section. Details are presented in response to suggestion nr 7.
- Add the gene pathway for the main gene to the result section. Describe very well the different possible mutations through a clear figure
We provided the entire pathway of tanshinone biosynthesis as Fig 6 at the end of 2.7 section. It includes mevalonic and MEP pathway as well as later stages of tanshinone biosynthesis process. We added several sentences to describe enzymes of pivotal role in particular pathway. They are located in Introduction section. Below is proposed text.
Plant hydroxymethylglutaryl-CoA reductase (HMGR) is recognized as the most important enzyme controlling the rate-limiting step in the MVA pathway (11). The HMGR is precisely regulated in plants at the level of transcription, post-transcription, translation and post-translation (12,13). Similar rate-limiting function i.e. the highest metabolite flux control coefficient in the MEP pathway indicates 1-deoxy-D-xylulose-5-phosphate synthase (DXS) (14). Therefore, the activity of DXS is precisely regulated at several post-translational levels (12,15). The significance of plastidial MEC enzyme is mediated predominantly by the product of its activity MECPD, considered as a retrograde signaling molecule affecting nuclear gene expression (16). Such hypothesis was verified by studies on mutants of the MEP pathway gene HDS, also known as ceh1 (constitutively expressing hydroperoxide lyase 1); these influence the expression of the enzyme HDS (1-hydroxy-2-methyl-2(E)butenyl4-diphosphate synthase) which converts MECPD into hydroxymethylbutenyl diphosphate (16-18). The mutants demonstrated higher MECPD concentrations, elevated salicylic acid (SA) level, greater resistance to infection by biotropic pathogens, and increased expression of a stress-inducible nuclear hydroperoxide lyase gene encoding a plastid-localized protein (16-18). Metabolic engineering approaches in S. miltiorrhiza hairy root cultures suggest that the enzyme geranylgeranyldiphosphate synthase (GGPPS) being active at later stages of tanshinone biosynthesis could more strongly induce the tanshinone accumulation rate than HMGR or DXS (19). The geranylgeranyldiphosphate (GGPP) produced by GGPPS is then used as a substrate by copalyl diphosphate synthase 1 (CPS1) and kaurene synthase-like 1 (KSL1) to the biosynthesis of miltiradiene, representing the complete but biologically inactive carbon structure of tanshinones (20). Further oxidative modification of miltiradiene skeleton introduced by numerous P450 cytochromes produce biologically active tanshinone molecules (20). These added sentences are marked in green.
- The discussion needs to be reframed as well to meet to journal requirements.
The discussion is reframed and some fragments are removed, added (as suggested other Reviewer) or reorganized to make the discussion more consistent. These added sentences are marked in green. Sentences to be removed are marked in red.
- In the materials and methods, the statical analysis is very weak and authors must improve it.
We added following sentences to describe statistical method used to find differences between tested samples.
The Wilcoxon signed-rank test was used to test samples before and after MeJa treatment with the aim of finding statistically significant differences. The Wilcoxon signed-rank test does not assume that the differences between paired samples are normally distributed. Therefore, the Wilcoxon signed-rank test has greater statistical power than Student's t-test and is more likely to produce a statistically significant result.
- The English language needs to be edited as well across the document.
The English language was edited, we provide also statement Attached pdf file) that the text was proofreaded by a native English speaker.
Reviewer 2 Report
Piotr Szymczyk and team aimed to clone 1559 bp long SmMEC gene promoter, 5’UTR and a short 5’ CDS DNA sequence. In silico analysis of promoter region reveald several cis-active elements. Authors have affirmed it by co-expression analysis with Arabidopsis thaliana. They have found that SmMEC gene was positively regulated by MeJa, as also confirmed by RT-PCR. Functional importance of MeJa (50-500 μM) in the regulation of SmMEC and tanshinone pathway exhibited the bi-phasic accumulation of total tanshinone in calluses (10-60 days) of S. miltiorrhiza; especially, CT and DHT were the prominent ones. Higher concentrations of MeJa (250-500 μM) decreasedthe callus GIF. Further, authors performed the in silico analysis of proximal promoter regions of MEP, MVA and other stages of tanshnone precursor biosynthesis, which verified that MeJa have role in the regulation of majority pathway genes.
Overall, this study is novel on the role of SmMEC gene promoter, its regulation by MeJa, and involvement in the production of tanshinone in calluses. Experiments are well planned, and executed in a logical way to achieve the objectives. Manuscript is nicely written, except some minor corrections in the figure, textual repetition/redundancy, answers to some queries. Please see the attached PDF, comments are annotated there.
Overall, I reccomend the acceptance of manuscript after minor corrections.

Author Response
We thank the Reviewer 2 for comments and the opportunity to improve the manuscript.
We carefully analyzed Reviewer 2 comments and provide following answers.
- Comments to Section 2.4
In addition to qPCR analysis of SmMEC gene, authors should perform Western blotting experiments to verify the activation of SmMEC protein.
Suggesting an evaluation of protein level changes after MeJa treatment could be a good point. However in our research we concentrated on quantitative RT-PCR analysis of SmMDS gene expression and other genetic aspects of its regulation., i.e. we concentrated at the transcription regulation level. We believe that such more proteomic-related analysis based on Western-blot could be interesting, however it could produce rather qualitative results and we should then probably validate it using costly quantitative proteomics studies. Moreover, there could not be a simple relationship between protein level indicated by Western-blot and enzymatic activity, so we should analyse also the enzyme activity before and after MeJa treatment. More importantly, there are also some technical hurdles; to perform such experiment we typically need to develop transgenic plant material transformed by a binary plasmid containing reference gene as for example gus fused to a flag to which are raised commercially available antibodies. The gus and fused flag should be controlled by SmMDS promoter, responding to MeJa treatment. Therefore to obtain even such qualitative Western-blot results we need to add several months of additional research, mainly to develop transgenic plant material.
We are aware of potential importance of such studies, proposed by reviewer 2, however we would like to ask the reviewer 2 not to perform this research, instead of this we prepared a text fragment that we suggest to add to the Discussion section. This fragment combines available research on S. miltiorrhiza showing positive relationship between SmMEC gene transcription rate and protein level after its induction by elicitation. We hope that such fragment at least partially will address the problem suggested by reviewer 2.
The addition of following text is proposed at the end of Discussion section:
The relationship between the transcription rate of MEP pathway enzymes and concentration of corresponding proteins was evaluated in S. miltiorhiza hairy root cultures elicited by 1 g L-1 yeast extract and 0.41 mM Ag+ (73). Quantitative LC-MS/MS analysis was applied to test precisely the protein concentration in the presented system. Obtained results suggest, that three MEP pathway proteins: 1-deoxy-D-xylulose-5-phosphate synthase (DXS), 2-C-methyl-D-erythritol 4-phosphate cytidylyltransferase (CMS), and 2C methyl-D-erithrytol 2,4-cyclodiphosphate synthase (MEC) were significantly upregulated after two weeks of elicitation. Extending the elicitation time to five weeks decreased or completely ceased the initial upregulation (73). The overexpression of MDS alone in S. miltiorrhiza hairy roots increases not only the concentration of the corresponding mRNA assayed by RT-PCR but also induces the total tanshinone concentration evaluated by HPLC (74). Presented data support the presence of putative relationship between increased MDS gene expression and upregulation of MDS enzyme level. Related proteomic studies were not performed on S. miltiorrhiza plant material treated by MeJa and could be addressed in future research.
- Changes in figure 2: we changed mM (in fact instead mM should be μM) to M, we also changed MeJA to MeJa.
- Changes in figure 3: we changed mM (in fact instead mM should be μM) to M, we also changed MeJA to MeJa.
- Changes in figure 4: we changed mM (in fact instead mM should be μM) to M, we also changed MeJA to MeJa.
- Did authors confirmed by measuring the cell death using flow cytometry or apoptosis analysis, or it is just an speculation, please clarify.
We did not perform such studies it is only our interpretation/speculation. We added sentence to explain it- marked in green.
For 500 μM MeJa treatment, even lower GIF values (<0.02) were observed after 20-40 days, with these values falling to slightly negative values -0.08 over longer periods, suggesting that cell divisions are outweighed by cell death processes. However, our interpretation is not supported by flow cytometry or apoptosis process analysis.
- Figure nr 5.
GIF increased at 50 and 100 mM after 60 days, any specific reason?
We could explain it in a following way. Both mentioned concentrations (50 and 100 μM) are lower among other two (250 and 500 μM) used in our study. It is visible that after 20 and 40 days these concentrations enable also relatively better callus growth rate as compared to higher concentrations. After 60 days these difference are particularly visible, however they could be observed also earlier, after 20 and 40 days.
The main reason is the fact that higher concentrations of MeJa shift plant cell metabolism too strongly towards secondary pathways- these pathways produce predominantly components of plant defence activity against different pathogens. Generally, the MeJa could just signal to the plant cell that it could be for example a prey of fungal or bacterial attack.
Moderate or low MeJa concentrations (50 and 100 μM) enable to maintain the relative balance between primary metabolic pathways, and mentioned secondary ones. So the plant cell has still enough energy carriers and organic substrates to promote cell division and biomass growth.
Highest MeJa concentration (250 and 500 μM) induces too strong imbalance between the primary and secondary metabolic routes, resulting in decreased concentration of available energy carriers and organic substrates, consumed by too powerfully induced secondary metabolite pathways. It could inhibits the cell division and biomass growth or even provoke cell death (negative growth index rates).
As correctly suggested reviewer 2, it is particularly visible at longer observations, i.e. after 60 days but occurs also earlier, after 20 and 40 days.
Other changes to Fig. 5. We changed mM to M.
- Reviewer 2 asked to transfer two fragments (green color) from section 2.7 to the Discussion and we did it.
- Two names indicating references (green color) in Discussion section were corrected to numbers.
- In section 4.6 and 4.8 we removed the doubled fragments.
Round 2
Reviewer 1 Report
The authors have well addressed all my query I have no more comments.
The manuscript can be accepted